# Memory and Markov Blankets

**DOI:** 10.3390/e23091105

**Published:** 2021-08-25

**Authors:** Thomas Parr, Lancelot Da Costa, Conor Heins, Maxwell James D. Ramstead, Karl J. Friston

**Affiliations:** 1Wellcome Centre for Human Neuroimaging, Queen Square Institute of Neurology, University College London, London WC1N 3AR, UK; l.da-costa@imperial.ac.uk (L.D.C.); maxwell.d.ramstead@gmail.com (M.J.D.R.); k.friston@ucl.ac.uk (K.J.F.); 2Department of Mathematics, Imperial College London, London SW7 2AZ, UK; 3Department of Collective Behaviour, Max Planck Institute of Animal Behavior, D-78457 Konstanz, Germany; conor.heins@gmail.com; 4Centre for the Advanced Study of Collective Behaviour, University of Konstanz, D-78457 Konstanz, Germany; 5Department of Biology, University of Konstanz, D-78457 Konstanz, Germany; 6Nested Minds Network, London EC4A 3TW, UK; 7Spatial Web Foundation, Los Angeles, CA 90016, USA; 8Division of Social and Transcultural Psychiatry, Department of Psychiatry, McGill University, Montreal, QC H3A 1A1, Canada

**Keywords:** Markov blanket, memory, conditional dependence, stochastic, density dynamics, Laplace assumption

## Abstract

In theoretical biology, we are often interested in random dynamical systems—like the brain—that appear to model their environments. This can be formalized by appealing to the existence of a (possibly non-equilibrium) steady state, whose density preserves a conditional independence between a biological entity and its surroundings. From this perspective, the conditioning set, or Markov blanket, induces a form of vicarious synchrony between creature and world—as if one were modelling the other. However, this results in an apparent paradox. If all conditional dependencies between a system and its surroundings depend upon the blanket, how do we account for the mnemonic capacity of living systems? It might appear that any shared dependence upon past blanket states violates the independence condition, as the variables on either side of the blanket now share information not available from the current blanket state. This paper aims to resolve this paradox, and to demonstrate that conditional independence does not preclude memory. Our argument rests upon drawing a distinction between the dependencies implied by a steady state density, and the density dynamics of the system conditioned upon its configuration at a previous time. The interesting question then becomes: What determines the length of time required for a stochastic system to ‘forget’ its initial conditions? We explore this question for an example system, whose steady state density possesses a Markov blanket, through simple numerical analyses. We conclude with a discussion of the relevance for memory in cognitive systems like us.

## 1. Introduction

This paper is about memory in systems that forget. The apparent paradox in this statement is the motivation for writing what follows. However, our first task is to highlight why forgetting is important. Simply put, if something were permanently changed by every perturbation it was exposed to, it would soon be unrecognizable as the thing it started out as. Living creatures are conspicuous examples of systems that maintain their characteristics over long periods of time, resisting the dissipative effects of accumulated perturbations. Perhaps the most famous articulation of this notion is the concept of homeostasis [1]—the idea that changes to our physiology are met with negative feedback mechanisms that act to restore our internal parameters to their pre-perturbation state. This just means that living systems have characteristic states such that, when sampled at random over time, they will be found in one of these characteristic states. In stochastic physics, these kinds of dynamics are often formulated with reference to a (steady state) probability distribution [2,3], to which the probability distribution of a random dynamical system returns after being perturbed.

In addition to forgetting their initial conditions, the systems we are interested in must exhibit a second property [4]. Specifically, we limit our scope to things that, under their steady state density, feature a separation between things internal and external to themselves. This is somewhat tautological, as it would be difficult to identify a thing if it were not separable from other things [5]. More formally, this implies conditional independence between internal and external states, given a third set of states, referred to as a Markov blanket [6], that acts as a boundary. For instance, photoreceptors in the retina act as a functional boundary between light sources and neural responses to light. By enforcing this separation, we ensure the memory in question is memory *about* something (i.e., the internal states forget something about the external states over time [7]). To restate the above, our focus is on systems with a steady state density, which factorizes such that internal states can be disambiguated from external states.

Having motivated the importance of forgetting, we now turn to the apparent problem with this construction. Namely, how can such a system remember? Specifically, if all dependencies between internal and external states are mediated via the blanket, internal states should not be able to retain information about external states when blanket states do not. To make this more concrete, consider the example above, in which retinal cell activity acts as the blanket separating the light reflected from some surface and the neural representation of that surface in the brain. Conditional independence between neural activity and the surface in question seems to imply that, when we close our eyes and eliminate the dependencies between the external and the blanket states, we should also destroy the dependencies between the internal and external states. In other words, we would forget about everything not currently in our field of view. 

We know from our normal, everyday, experience that we can remember things that we cannot currently see. In addition an important body of work in neurobiology depends upon the phenomenon of ‘delay-period’ activity, in which persistent increases in neural firing rates are observed in the time-period between a stimulus disappearing and a creature acting upon what it had seen [8]. So how do we reconcile the special type of independence evinced by systems that maintain a steady state separation—between their inside and outside—with the fact that such systems are capable of memory? 

The answer is that the steady state is where we end up, when our initial conditions have been forgotten. More precisely, if we have some knowledge of the variables comprising a stochastic system, this information is useful in determining the distribution of these variables we would anticipate a short time in the future. However, as time progresses, the accumulated fluctuations render these initial values less informative. Given a sufficiently long time, those values become completely uninformative, in the sense that all initial conditions would have resulted in the same distribution. This distribution is the steady state density. However, as soon as we say that something has been observed (e.g., photoreceptor activation from light reflected from a surface), we are implicitly imposing new initial conditions take time to be forgotten. During that time, the evolving probability density—describing the components of a Markov blanket—do not feature the conditional independencies of the steady state density that define the Markov blanket. 

In the sections that follow, we rehearse the argument above, both analytically and through numerical simulation. We exploit a Laplace assumption [9] that assumes a (locally) quadratic form for the log probability density (i.e., that it is approximately described by a multivariate Gaussian density). Using the Laplace assumption, we formulate the evolution of a system whose steady state density possesses a Markov blanket, in terms of the mode and variance of the evolving density, and highlight several ways in which memory could be measured following a perturbation. This lets us examine the influence of different aspects of the system’s dynamics on these measures. Finally, we consider how this might generalize to non-Gaussian steady states, including those described by categorical probability distributions. We demonstrate that no paradox is entailed by the mnemonic capacities of systems whose steady-state density is endowed with a Markov blanket.

## 2. Markov Blankets and Density Dynamics

This section offers a formal definition of a Markov blanket [6,10], and sets out the density dynamics that ensure a steady state consistent with the requisite conditional (in)dependencies. A Markov blanket (*b*) is defined in terms of two other sets of variables, here referred to as internal (*µ*) and external (*η*) states. The blanket is the set of variables that render the internal and external states conditionally independent of one another. Conditional independence can be expressed in terms of the following factorization of a probability distribution or density:(1)$$\left(\mu ⊥\eta \right)|b\Leftrightarrow p(\mu ,\eta |b)=p(\mu |b)p(\eta |b).$$Expressed more intuitively: If we knew everything about the blanket, knowing the external states would tell us nothing that we did not already know about internal states, and vice versa. 

A common example of a Markov blanket is found in the context of a Markov process [11,12], in which the present state of the world acts as the blanket that separates the past from the future. However, conditional independencies of this sort can be found in many real-world systems [6,13,14]. In biology, as highlighted in the introduction, the state of a sensory receptor acts as a blanket that insulates the activity of the nervous system from the external stimulus that acts upon the receptor.

An important aspect of biological systems is that they have a temporal dependence, in the sense that the conditional independencies arise from an underlying dynamical system with an attracting set of states (described probabilistically with the steady-state density). This is particularly important for questions about memory, which presupposes persistence over time. We can link a system’s dynamics and its statistical dependencies by appealing to stochastic differential equations and their associated density dynamics [11]: (2)$$\begin{array}{c} x=\left(\mu ,b,\eta \right)\\ \dot{x}=f(x)+\omega (x)\Leftrightarrow \\ \partial_{\tau }p(x,\tau )=\nabla_{x}\cdot \left(\nabla_{x}\cdot \left(\Gamma (x)p(x,\tau )\right)\right)-\nabla_{x}\cdot \left(f(x)p(x,\tau )\right) \end{array}$$

Equation (2) shows two alternative descriptions of a random dynamical system. The first is a stochastic differential equation that describes the rate of change of a variable *x* (here comprising all constituents of a Markov blanketed system) as a function of a deterministic flow (*f*(*x*)) and normally distributed (white noise) fluctuations (*ω*(*x*)) with covariance (2Γ). As *ω*(*x*) has units of velocity, we can relate it to the speed of fluctuations. This means that, when Γ(*x*) is large, there are more fast fluctuations than when Γ(*x*) is small. When the amplitude of random fluctuations is large, the probability density over future states disperses rapidly—and the initial conditions are, effectively, forgotten more quickly. The evolution of the probability density describing *x* is shown in the final line. This deterministic partial differential equation is known as a Fokker–Planck equation [11,15]. The notation *p*(*x*,*τ*) inherits from the conventions used in [3], and should be read as the probability density of a stochastic process *X* evaluated at *x* at time *τ*.

As alluded to above, our interest is in systems that possess a steady state density. Operationally, this simply means that the probability density stops changing. Setting the rate of change of the probability density to zero and solving for the associated flow gives us the following form:(3)$${\left.\partial_{\tau }p(x,\tau )\right|}_{\tau =\infty }=0\Leftrightarrow \left\{\begin{array}{l} f(x)=\left(Q(x)-\Gamma (x)\right)\nabla_{x}ℑ(x)+\nabla_{x}\cdot (\Gamma (x)+Q(x))\\ p(x,\infty )\propto e^{-ℑ(x)}\\ \nabla_{x}\cdot \left(\nabla_{x}\cdot \left(Q(x)p(x,\infty )\right)\right)=0 \end{array}\right.$$

Equation (3) says that the flow of a random variable *x* can be expressed in terms of the gradient of a potential function (ℑ) that corresponds to the negative logarithm of the steady state density at each point in the state space. The flow is decomposed into two parts [10,16,17]: the flow along the contours of the potential (*Q*) and the flow down the potential gradients (Γ). The gradient flow counters the dispersive effect of random fluctuations (Equation (2)) to maintain the steady state density. The final term on the first line of the right-hand side of Equation (3) corrects for the effect of variations in the covariance of fluctuations as a function of *x*. This uses the convention for matrix divergences advocated in [18]. However, in what follows, we will assume the covariance of the fluctuations (2Γ) and the solenoidal flow (*Q*) vary sufficiently slowly, with respect to *x*, that they can be considered constant. Taken in combination with the Laplace assumption, the resulting flows become linear functions. In addition to simplifying the treatment that follows, this implies that the dynamics in Equation (2) can be interpreted equivalently under Itô, Stratonovich, or A-type conventions [19] for stochastic differential equations. For an example where this assumption is relaxed, please see [20], which describes a parameterization of the quantities in Equation (3) that can be fit to approximate the behavior of systems like the Lorenz attractor.

The above formalizes the first condition in the introduction: The forgetting of initial conditions. This is because, given sufficient time, regardless of the initial condition, the probability density will converge to its steady state. The second condition we highlighted was that the steady state should factorize in accordance with Equation (1), such that it contains a Markov blanket. Taken together, these conditions mean that internal states must eventually forget about external states, so that the two become conditionally independent given the blanket at steady state.

We turn now to the apparent problem. At a first pass, it might seem that these two conditions preclude any capacity for memory. If this were true, it would suggest that these dynamics would be inappropriate as a description of cognitive systems with mnemonic abilities (like us). However, it is worth thinking carefully about what we mean when we talk about memory. Normally, a memory is defined in relation to an observation that occurs at a particular time. For instance, if we were asked to remember a phone number, we are first told the number—generating an auditory sensation (blanket state)—after which the internal states in our brains must maintain some information about the external state (the phone number) that caused that auditory impression. Implicit in this is the idea that we are conditioning our probability density on some initial condition so, by definition, the density is not at steady state. By ‘initial conditions’, we mean the marginal density over states at the start of some epoch (usually set to be a Dirac delta function of one or more states). Using *b*_0_ to indicate the blanket state at some reference time, and *τ* to represent the time that has elapsed since that reference, we have:(4)$$p\left(\eta ,\mu ,b\right)=\underset{\tau \to \infty }{\lim}p\left(\eta ,\mu ,b,\tau |b_{0}\right)$$

The left side of Equation (4)—the steady state density—factorizes as in Equation (1). However, the internal and external states are both conditionally dependent upon the initial blanket state until the evolving density has converged upon its steady state, they may not be conditionally independent of one another given only the present blanket state. Given sufficient time, accumulated fluctuations will render all parts of the blanket independent of the initial state, but in the intervening time, the internal and external states may appear dependent upon one another. In other words, instantaneous statistical dependency between internal and external states may derive from a ‘common past’ (i.e., some initial condition), rather than being solely mediated by contemporaneous blanket states. The implication is that during this relaxation back to steady state, the densities evolving towards a steady state density with a Markov blanket are not themselves subject to the Markov blanket conditions. Consequently, the internal states can remember something about the external states.

The above establishes that the presence of a Markov blanket at steady state does not preclude mnemonic processing. However, it does raise an interesting question. What determines the duration of a memory? In Section 3, we describe a stochastic system that conforms to the conditions above. We use this to illustrate the point made in Equation (4) and demonstrate how we might measure the persistence of a memory. 

## 3. Density Dynamics under the Laplace Assumption

To make the relatively abstract discussion in Section 2 a little more concrete, it is useful to commit to an example. For simplicity, we will deal with a system that conforms to a Laplace assumption (i.e., that has a potential function with a quadratic form). Given a quadratic form, the Markov blanket condition is simple to enforce. The Hessian of the potential Π expresses conditional dependencies with non-zero elements. As such, if we set the elements corresponding to the interaction between internal and external states to zero, we ensure a Markov blanket at steady state:(5)$$\begin{array}{l} ℑ(x)=\frac{1}{2}\left(x-\vartheta \right)\cdot \left(x-\vartheta \right)\\ \Pi_{\eta \mu }=0 \end{array}$$

In Equation (5), $\Pi_{\eta \mu }$ is the element of the Hessian matrix of the potential function (i.e., negative log steady state density) corresponding to the internal and external states. Table 1 sets out the values used for the numerical simulations that follow. Unless otherwise specified, we set *γ* = *κ* = *θ* = 1. Figure 1 shows a simulation of the stochastic differential equation resulting from Equations (2) and (3), with the potential from Equation (5). This was simulated 512 times, with the upper and middle plots showing the state of each simulation by *τ* = 16. The lower plot shows the mean values of each variable at each time. Each simulation was initialized at *x* = (1,1,1). As these simulations show, the average trajectories are damped oscillations. This is to be expected under the decomposition in Equation (3), as *Q* induces an orbit around the potential, while Γ damps these oscillations and brings the system towards its most likely value.

The distributions shown in the upper and middle plots of Figure 1 are non-spherical. This means each pair of states covaries. However, this does not mean that all are conditionally dependent upon one another. The covariance between internal and external states is a consequence of the conditional dependencies each shares with the blanket states. To fully understand how these dependencies vary with time, we can move from the stochastic dynamics of Figure 1 to the density dynamics depicted in Figure 2. 

To get to these density dynamics, we need a parametric form for the probability density. Under the Laplace assumption and the assumption that the matrix fields Γ and *Q* do not vary in space, this is given by the multivariate normal distribution [21]. Substituting this into the Fokker–Planck equation, with the quadratic potential function in Equation (5), we have [9]:(6)$$\begin{array}{l} p(x,\tau )=N\left(\xi ,\Sigma \right)\\ \;\;\dot{\xi }=-\left(\Gamma -Q\right)\Pi \left(\xi -\vartheta \right)\\ \;\;\dot{\Sigma }=2\Gamma -\Sigma \Pi {\left(\Gamma -Q\right)}^{T}-\left(\Gamma -Q\right)\Pi \Sigma \end{array}$$

This can be re-expressed—noting that the final line is a Sylvester equation—as:(7)$$\begin{array}{l} \left[\begin{array}{l} \dot{\xi }\\ vec\left(\dot{\Sigma }\right) \end{array}\right]=\left[\begin{array}{l} \left(\Gamma -Q\right)\Pi \vartheta \\ 2vec\left(\Gamma \right) \end{array}\right]\\ \;\;\;\;\;-\left[\begin{array}{cc} \left(\Gamma -Q\right)\Pi & 0\\ 0 & \left(\Gamma -Q\right)\Pi ⊗I+I⊗\left(\Gamma -Q\right)\Pi \end{array}\right]\left[\begin{array}{l} \xi \\ vec\left(\Sigma \right) \end{array}\right] \end{array}$$

The zero elements of the Jacobian (the matrix on the second line of Equation (7)) represent zero block matrices. The ‘vec’ notation in Equation (7) is the matrix vectorization operation. Figure 2 shows what happens when we numerically integrate Equation (7). The upper plots show the steady state obtained at *τ* = 16, which is consistent with the distribution seen in the stochastic simulations of Figure 1. The density dynamics of Figure 2 are the limiting case of the stochastic system in Figure 1 as the number of simulations conducted is increased. The convergence evident in Figure 2 illustrates that *τ* = 16 is sufficient for the evolving density to reach a steady state. The trajectory of the mode shown in the middle plot of Figure 2 closely resembles the averaged trajectories in Figure 1. However, the key message of Figure 2 is in the lower plot, which shows how the elements of the precision (inverse covariance) matrix evolve over time. When an element of the precision matrix is non-zero, it means the variables associated with the column and row are conditionally dependent upon one another. By the end of the simulation, the elements of the precision matrix converge upon those of the Hessian matrix of the steady state potential. On their way to these values, the internal and external states acquire a transient conditional dependency. This implies a synchronization beyond that given by the (current) blanket state and can be regarded as a numerical realization of Equation (4).

## 4. Measures of Memory

Having demonstrated that the presence of a Markov blanket at steady state does not preclude conditional dependencies across the blanket during the evolution towards steady state, a new question presents itself: What determines the persistence of these dependences following a perturbation, and how should we go about measuring this? The period during which the internal-external element of the precision matrix is non-zero represents one measure we could adopt, but there are other ways in which we can look at memory. 

Some of these measures can be developed from the Kullback Leibler (KL) divergence [22]. Under the Laplace assumption, this has a simple form [23]:(8)$$\begin{array}{ll} D_{KL}\left[N(\alpha ,A)∥N(\beta ,B)\right] & =\frac{1}{2}\left(tr\left(B^{-1}A\right)+(\alpha -\beta )\cdot B^{-1}(\alpha -\beta )\right.\\ & \left.-\dim(\alpha )+\ln\left(\det(B)\right)-\ln\left(\det(A)\right)\right) \end{array}$$

The KL divergence is an asymmetrical measure of how different two probability densities are from one another. It measures the average difference between two log probability densities. There are two ways we can use this to examine the density dynamics shown in Figure 2. The first is to measure the mutual information between the internal and external states given blanket states. Under the Laplace assumption, this takes the form:(9)$$\begin{array}{ll} I\left[p(\mu ,\tau |b);p(\eta ,\tau |b)\right] & ≜D_{KL}\left[p(\mu ,\eta ,\tau |b)∥p(\mu ,\tau |b)p(\eta ,\tau |b)\right]\\ & =\frac{1}{2}\ln\left(\frac{\det({ϒ}_{\eta \eta })}{\det({ϒ}_{\eta \eta }-{ϒ}_{\eta \mu }{ϒ}_{\mu \mu }^{-1}{ϒ}_{\eta \mu }^{T})}\right)\\ \Upsilon & ≜\left[\begin{array}{cc} \Sigma_{\mu \mu } & \Sigma_{\mu \eta }\\ \Sigma_{\mu \eta }^{T} & \Sigma_{\eta \eta } \end{array}\right]-\left[\begin{array}{l} \Sigma_{\mu b}\\ \Sigma_{\eta b} \end{array}\right]\Sigma_{bb}^{-1}{\left[\begin{array}{l} \Sigma_{\mu b}\\ \Sigma_{\eta b} \end{array}\right]}^{T} \end{array}$$

The conditional covariance Υ is the Schur complement of the covariance of the blanket states in Σ [22,24]. For reference, Table 2 reproduces some standard results useful in deriving Equation (9). The mutual information shown in Equation (9) tells us how informative internal states are about external states (and vice versa) once we have discounted the shared information due to the blanket states. More formally, it is the divergence between a joint density (including all dependencies) and the product of two marginal densities (in which dependencies between internal and external have been omitted). Once the Markov blanket condition has been satisfied, this mutual information will be zero. The period for which it is non-zero is; therefore, another measure of the time during which internal states ‘remember’ external states.

In addition to thinking about the difference between joint and marginal densities, we can also think about divergences between the same densities at different times. Here, there is a point of contact with an important measure used to characterize itinerancy in non-equilibrium systems [26,27,28]. This approach borrows from information geometry [29,30], which applies methods from differential geometry to probability theory. Specifically, information geometry works with statistical manifolds that represent spaces of probability densities. A central concept in this setting is information length, which is a measure of the distance between densities. As the distance from one point to another is the same as the distance from the second point to the first, the asymmetry of a KL divergence renders it inappropriate as a distance measure. However, for very small changes in density, the divergence is approximately symmetric. The leading term in a Taylor expansion of the KL divergence is known as the Fisher information metric tensor, which provides us with a (symmetric) measure of distance between probability distributions [31,32,33]: (10)$$\begin{array}{ll} \frac{1}{2}dl{\left(\tau \right)}^{2} & =\underset{0}{\underset{︸}{D_{KL}\left[p(x,\tau )∥p(x,\tau )\right]}}+d\tau \underset{0}{\underset{︸}{{\left.\partial_{\tau }D_{KL}\left[p(x,\tau +d\tau )∥p(x,\tau )\right]\right|}_{d\tau =0}}}\\ & +\frac{1}{2}d\tau^{2}{\left.\partial_{\tau }^{2}D_{KL}\left[p(x,\tau +d\tau )∥p(x,\tau )\right]\right|}_{d\tau =0}\\ & \Rightarrow \\ dl(\tau ) & \approx \sqrt{2D_{KL}\left[p(x,\tau +d\tau )||p(x,\tau )\right]} \end{array}$$

Consequently, we can measure the length of the path the evolving density takes along the statistical manifold by taking the square root of twice the KL divergence for each time interval. We can do this both for the joint density and for the marginal density for the blanket states. The difference between these information lengths can be regarded as the distance travelled by the system that is not attributable to the change in the marginal density of blanket states. 

Given the linearity of Equation (7), one further measure we can consider is the reciprocals of the (real parts of the) eigenvalues of the Jacobian for the covariance matrix. These play the role of temporal decay constants. The longer these time constants, the greater the memory span of the system. However, these are less specific than the measures above, and do not distinguish between blanket-dependent and blanket-independent persistence in synchronization between the internal and external states. 

## 5. Numerical Simulations

Unpacking Equation (6), the instantaneous rate of change of the element of the precision matrix linking internal and external states is:(11)$$\begin{array}{ll} \Phi & ≜\Sigma^{-1}\\ \dot{\Phi } & =\Pi {\left(\Gamma -Q\right)}^{T}\Phi +\Phi \left(\Gamma -Q\right)\Pi -2\Phi \Gamma \Phi \\ {\dot{\Phi }}_{\mu \eta } & =\Gamma_{\mu \mu }\left(\Pi_{\mu \mu }-2\Phi_{\mu \mu }\right)\Phi_{\mu \eta }\\ & +\Gamma_{\eta \eta }\left(\Pi_{\eta \eta }-2\Phi_{\eta \eta }\right)\Phi_{\mu \eta }\\ & +\Gamma_{bb}\left(\Pi_{\mu b}\Phi_{b\eta }+\Pi_{\eta b}\Phi_{b\mu }-2\Phi_{\mu b}\Phi_{b\eta }\right)\\ & +Q_{\mu b}\left(\Pi_{\mu \mu }\Phi_{b\eta }-\Pi_{b\eta }\Phi_{\mu \mu }-\Pi_{\mu b}\Phi_{\mu \eta }\right)\\ & +Q_{\eta b}\left(\Pi_{\eta \eta }\Phi_{\mu b}-\Pi_{\mu b}\Phi_{\eta \eta }-\Pi_{\eta b}\Phi_{\mu \eta }\right) \end{array}$$

If we substitute the parameterization from Table 1, Equation (11) becomes:(12)$$\begin{array}{ll} {\dot{\Phi }}_{\mu \eta } & =\gamma \underset{\mathrm{Dissipation}\;\mathrm{due}\;\mathrm{to}\;\mathrm{internal}\;\mathrm{and}\;\mathrm{external}\;\mathrm{states}}{\underset{︸}{\left(\kappa^{-1}-\frac{1}{2}\Phi_{\mu \mu }-\frac{1}{2}\Phi_{\eta \eta }\right)\Phi_{\mu \eta }}}\\ & +\underset{\mathrm{Dissipation}\;\mathrm{due}\;\mathrm{to}\;\mathrm{blanket}\;\mathrm{states}}{\underset{︸}{\left(\kappa^{-\frac{1}{2}}\Phi_{b\eta }+\kappa^{-\frac{1}{2}}\Phi_{b\mu }-2\Phi_{\mu b}\Phi_{b\eta }\right)}}\\ & +\theta \underset{\mathrm{Solenoidal}\;\mathrm{coupling}\;\mathrm{to}\;\mathrm{blanket}\;\mathrm{states}}{\underset{︸}{\left(2\kappa^{-1}\Phi_{b\mu }-2\kappa^{-1}\Phi_{b\eta }+\kappa^{-\frac{1}{2}}\Phi_{\mu \mu }-\kappa^{-\frac{1}{2}}\Phi_{\eta \eta }\right)}} \end{array}$$

This discloses the way in which each parameter might influence the rate of decay of the synchronization between the internal and external states. For instance, the dissipative flow mediated by *γ* only becomes relevant when the internal and external states are conditionally dependent upon one another. This implies that the induction of a conditional dependence must occur via the dissipation of blanket states or the solenoidal coupling to these states. When at steady state, all the terms involving the solenoidal coupling cancel each other out. The form of the steady state density interacts with both the solenoidal and dissipative flows. However, the rate of change of this element of the precision matrix depends upon the other elements of the precision matrix, which will themselves evolve over time. 

To better understand the influences on memory, we will use a series of numerical simulations to examine the influences of each of these parameters on the measures of memory outlined in Section 4. To do this, we initialize the system with a density consistent with steady state but with one important difference. This is to set the precision associated with the blanket state of the initial density to be very high (*e*^4^), as if we had just observed this state—and found it to be at its expected value. This imposes an initial condition on the system that breaks the conditional independencies required for a Markov blanket, until that initial condition is forgotten.

The qualitative relationship between the parameterizations of Table 1 and the measures outlined in Section 4 is more important for the central message of this paper than the specific quantitative differences, which pertain only to the specific Ornstein–Uhlenbeck process used here.

### 5.1. The Diffusion Tensor

First, we examine the role of the diffusion tensor (Γ), parameterized by *γ*, which acts as a rate constant for the dissipative component of the flow. As we are interested in the synchronization of internal and external states, we have assumed a symmetric parameterization, such that both internal and external flows are manipulated together. Figure 3 reports the results of a series of simulations at different values of *γ*. We might anticipate that systems with fast fluctuations should very rapidly forget their initial conditions, such that large *γ* should lead to rapid forgetting. Some readings of quantum mechanics depend upon this rapid evolution of the density to steady state [5]. The results in Figure 3 appear to endorse this intuition. When *γ* is small, the time constants are slower, the information length travelled is longer, and both the mutual information and precision have larger absolute values later into the simulation. 

However, Figure 3 suggests an additional nuance. Note the pattern of synchronization and desynchronization in the precision and mutual information plots. The periodicity of this pattern is not influenced by changes in the diffusion tensor—which only influences the relative size of each peak. Another way of putting this is that the time at which the desynchronization between each peak occurs is not changed by the diffusion tensor. This highlights the importance of the solenoidal flow, which we turn to next. 

The other aspect of this is that changes in external and internal states can themselves lead to changes in the blanket states, following the initial perturbation. This means the blanket states go through a series of probabilistic configurations on their way back to steady state that induce further changes in the external and internal states. Conceptually, this can be thought of as a form of memory in which our sensations cause changes in our internal (neural) states, which then cause actions that themselves cause a transient re-synchronization between the internal and external world, echoing the influences of the original observation. This speaks to the symmetry of inference and memory between a creature and its environment [34], as transient synchronization occurs both when the outside world changes (or is changed) to conform to neural states and when neural states change to represent the external world.

### 5.2. Solenoidal Flow

Intuitively, we might expect a greater memory span for a system that takes a more convoluted path back to steady state following perturbation. Itinerancy of this sort can be induced by giving the solenoidal flow a larger part to play. The (linear) association between solenoidal flow and the information length of the path to steady state has been convincingly demonstrated for stochastic systems that conform to a Laplace assumption [28]. Figure 4 reproduces this result, in addition to the other measures from Figure 3, by varying *θ*. However, the interpretation of these simulation results requires some care.

Although augmenting the solenoidal flow increases the information length, as expected, it has very little influence over the time constants. The implication is that, despite the path being longer, the speed with which it is travelled increases approximately proportionally. In a sense, it is surprising that solenoidal flow has any influence at all over memory span, as its action is around the contours of the non-equilibrium steady state (NESS) density—unlike the dissipative flow towards regions of high probability. Although counterintuitive, we can make sense of this by considering what it means to move along the gradients of the NESS when away from the mode. This increases the covariance between variables that covary under the NESS density, but show limited covariance immediately following an observation. This explains the small decrease in time constant when the solenoidal coupling is increased—an extensively studied phenomenon in the sampling literature [35,36]. The effect of the change in time constants can be observed in the mutual information plot, where the difference in amplitude between the first two peaks is greater when the solenoidal coupling is greater (i.e., when the time constant is faster) than when solenoidal coupling is weaker. Under larger values of *γ*, the rate of decrement in each peak for the mutual information is much faster. For very small *γ*, this decrement is much slower.

In addition to the slowing of the time constants, decreases in the strength of solenoidal coupling increases the period of the (damped) oscillations visible in the upper two plots, so that each individual period of synchronization lasts longer. However, this slowing also means that the peaks occur later. Given the damping effect of the dissipative flow, later peaks mean smaller amplitude peaks. This suggests that there is a trade-off in solenoidal coupling for memory capacity. Although a memory may last longer with reduced solenoidal coupling, its amplitude (i.e., the amount of information carried) is sacrificed. An implication of this is that there is an interaction between the solenoidal and dissipative flows, in the sense that the greater the dissipative flow rate, the smaller the amplitude of the solenoidal path, and the shorter the information length.

### 5.3. Variance at NESS

The final parameter to examine is the *κ* variable that parameterizes the precision at steady state. First, it is worth unpacking the significance of the choice of parameterization shown in Table 1. This is chosen so that we can make a very specific manipulation to the form of the NESS density. Specifically, it lets us dissociate two different sorts of variance:(13)$$\begin{array}{c} p\left(\mu \right)=N\left(\lambda_{\mu },\Sigma_{\mu }\right)\\ p\left(\eta \right)=N\left(\lambda_{\eta },\Sigma_{\eta }\right)\\ p\left(b|\mu \right)=N\left(\lambda_{b|\mu },\Sigma_{b|\mu }\right)\\ \;p\left(b|\eta \right)=N\left(\lambda_{b|\eta },\Sigma_{b|\eta }\right)\\ \Sigma_{\mu }=\Sigma_{\eta }=\frac{3}{4}\kappa \\ \Sigma_{b|\mu }=\Sigma_{b|\eta }=\frac{2}{3} \end{array}$$

In Equation (13), we see that the variance of the marginal densities of the internal and external states scales with *κ*, but that the variances of the conditional densities for the blanket states remain constant. From a Bayesian perspective, we can think of *κ* as modulating the relative variances of a prior (marginal) and likelihood (conditional) density. Another way of putting this is that it is a hyperparameter of the model implied by the steady state density and plays a different role to the *γ* and *θ* parameters of the flow operator. With this in mind, we can start to interpret the numerical results reported in Figure 5. The first thing to note is that the time constants shown in the lower plot scale with the variance of the marginal densities. This is to be expected, as both the solenoidal and dissipative flows depend upon the gradients of the log NESS density, which become shallower as *κ* increases.

Given the influence of *κ* on both parts of the flow, evident in Equation (12), it is unsurprising that increasing *κ* not only slows the period of the damped oscillations but increases the damping itself. This has an interesting implication for the information length, for which there is a rapid increase when the marginals have a tight variance, but result in a shorter overall information length travelled, as the slower dynamics with a wider variance NESS end up overtaking.

An intuitive way of looking at the results in Figure 5 is that a tight prior belief about how the world should be results in a vigorous response to it being otherwise. This accounts for the high early peak in mutual information. As this response quickly restores the world to a state that complies with prior beliefs, the perturbation is quickly forgotten. In contrast, a more permissive prior belief allows for a more sustained deviation from steady state. During this deviation, internal and external states can maintain a conditional dependence upon one another.

In neuronal systems, the kind of dynamics outlined here could be associated with distributions of neuronal membrane potentials, firing rates, or synaptic efficacies and continuous external variables, such as luminance or joint angles. Some examples of systems based upon (local) Laplace approximations—treating the steady state density as if it were an internal model—include models of songbirds in communication with one another [37], generation of handwriting [38], and motor control [39].

## 6. Categorial Distributions

In the previous section, we examined the influence of various parameters on transient conditional dependencies between states which, under their steady state density, are on opposite sides of a Markov blanket. However, this analysis was limited to systems that can be approximated under a Laplace assumption (i.e., systems defined in terms of continuous random variables close to the mode of some steady state density). In this section, we briefly unpack what the same analysis might look like in the domain of categorical probability distributions. 

### 6.1. Master Equations and Markov Blankets

The first step we took for the Laplace system above was to find the dynamics of the probability distributions that converge upon a distribution with a Markov blanket at steady state. We can do the same thing for a categorical distribution using a Master equation:(14)$$\begin{array}{c} s=(\mu ,\eta ,b)\\ P(s,\tau )=Cat(s(\tau ))\\ vec(\dot{s}(\tau ))=Lvec(s(\tau )) \end{array}$$

Equation (14) expresses the rate of change of **s**, a three-dimensional array (i.e., an array with dimension *N_µ_* × *N_b_* × *N_η_* where *N_i_* are the number of possible values the *i*-th state can take) encoding the probabilities that the internal, external, and blanket states occupy some combination of countable states. The transition rate matrix **L** may be re-expressed in terms of a steady state distribution following the approach of [40]:(15)$$\begin{array}{c} L=\Omega \Lambda \\ \Lambda ≜diag{\left(vec(s(\infty ))\right)}^{-1}\\ s_{\mu \eta b}(\infty )=s_{\mu |b}(\infty )s_{\eta |b}(\infty )s_{b}(\infty ) \end{array}$$where **Ω** is defined in Equation (16). In analogy with the dynamical decomposition in continuous state spaces, Equation (15) can be expressed in terms of a flow that preserves detailed balance (**Γ**) and a flow that does not (**Q**):(16)$$\begin{array}{c} vec(\dot{s}(\tau ))=(Q-\Gamma )\Lambda vec(s(\tau ))\\ \sum_{j}\Gamma_{ij}=\sum_{i}Q_{ij}=0\\ \Omega =Q-\Gamma \\ \Gamma =-\frac{1}{2}(\Omega +\Omega^{T})\\ Q=+\frac{1}{2}(\Omega -\Omega^{T}) \end{array}$$

Equation (16) has a form closely resembling the evolution of the expectation in Equation (6). Although we exploit the analogy in the form of the equations, it should be stressed that the analogy is inexact and should not be overinterpreted. For example, if were to attempt to scale the **Γ** and **Q** terms independently of one another, as we did with the dissipative and solenoidal flows in the continuous domain, we could end up with negative elements in the off-diagonal terms of the probability transition rate matrix. This in turn could lead to negative probabilities. As such, we must take care in the manipulations we choose. 

### 6.2. Simulations

Figure 6 shows a simulation of a system described by Equation (16), parameterized as in Table 3. The parameters of Table 3 repurpose the same symbols as in Table 1, in line with the loose analogy between Equations (6) and (16). As in the simulations for Figure 3, Figure 4 and Figure 5, we initialize with a distribution consistent with steady state except for a precise distribution centered on the most likely blanket state. Figure 6 shows the expected blanket states rapidly returning to the expectation under the steady state density, causing subtle transient changes to the internal and external state expectations on the way.

Except for the precision matrix, which has a specific role in normal distributions, the measures from Figure 3, Figure 4 and Figure 5 are agnostic to the family of distributions describing the associated random variables. This means they can also be used to examine the categorical system. Figure 7 reports a non-exhaustive numerical analysis of the system from Equation (16). The main observation to take away from Figure 7 is that, for a range of parameter values, the transient conditional dependencies we encountered in Section 6 emerge once we impose initial conditions upon the blanket states. This generalizes the central message of this paper—that there is no contradiction between Markov blankets and memory—to categorical probabilistic dynamics.

In addition, there are several interesting parallels and differences between the continuous and categorical systems. First, as the dissipative part of the flow increases in rate, the time for the mutual information to return to zero, and for the information length to converge, decreases. Consistent with this is the decrease in the slowest time constant. Comparison with Figure 3 shows similar qualitative results. Differences between the categorical and continuous systems are more apparent in the distinction between the role of solenoidal flows in continuous and categorical systems. In Figure 7, changing *θ* does little to measures reported. Although this is also true for the time constants in the continuous domain, (see Figure 4), increasing the solenoidal flow rate in the continuous system has a profound influence over the path pursued.

The influence of the shape of the NESS density is shown on the right of Figure 7. The *κ* parameter here is a slightly coarser manipulation than the analogous plot in Figure 5. Here, it plays the role of an inverse temperature parameter, which can be loosely thought of as an inverse variance. It is applied to both the marginal over the blanket and the two conditional distributions. Given the broad influence of this parameter, it is unsurprising that its effects on the measures in Figure 7 are also widespread. Some time constants increase as the inverse temperature increases, while others decrease. At high values, there is a sharp increase in the mutual information that rapidly decays, compared to a smaller, broader, and later peak for the lower values.

Clearly the parameterization in Table 3 is one of many possible configurations for the dynamics of a categorical stochastic system. However, it serves to illustrate three points. First, the conditional mutual information, information length, and eigenvalues of the Jacobian can be used as measures of different aspects of memory, both in continuous and categorical systems. Second, like continuous systems, categorical systems undergo transient conditional dependencies across a Markov blanket following perturbation (i.e., imposition of initial conditions). Third, the form of that transient synchronization is a function of the steady state distribution, and of flow operators that determine the path to that steady state.

Applications of categorical distributions in neurobiology might include representations of alternative objects, faces, or locations. In addition, these distributions feature heavily in models of decision making [41], where creatures must select between alternative plans of action.

## 7. Discussion

### 7.1. Memory in Markov Blanketed Systems

We started this paper by outlining the importance of a Markov blanket. In brief, a blanket ensures statistical separation between two sets of variables (internal and external states). Without this separation, we would not be able to refer to one part of a system holding memories *about* another part of that system, as we would have a homogenous system with no notion of parts. For this separation to persist through time, there must be a steady state distribution to which the probability distribution evolves. However, there is an apparent contradiction here. If cognitive systems, like us, with mnemonic capacity can be described in terms of Markov blankets, the implication is that internal and external states should only carry information about one another when this information is mediated by the blanket states. This appears to preclude nearly all forms of memory, as conditioning upon the past implies there is a source of shared information between internal and external states in addition to the blanket states.

We have seen that this contradiction is specious, and rests upon a misconception of a steady state. A system cannot *be* at steady-state—it only *has* a steady-state density. This density is the solution to the Fokker–Planck or master equation describing density dynamics. In other words, the system will evolve to the steady-state density from any initial conditions as it is rolled out (or pushed forward) into the future. This means that, as soon as we condition upon some past state, we have imposed new initial conditions under which the density is no longer the steady state density. As such, the Markov blanket condition does not apply until these conditions have been forgotten.

Under the Laplace assumption, we saw the emergence of a transient synchronization between internal and external states, on imposing initial conditions on blanket states. This means that the probabilistic configurations we must move through, between observing a blanket state and reaching a distribution consistent with steady state, include configurations in which the internal and external states conditionally depend upon one another.

This memory can be quantified using measures derived from information theory, information geometry, and from dynamical systems theory. The time course of each of these measures varies systematically, for the specific system in question, with manipulations to the different parts of a flow operator: Including solenoidal and dissipative gradient flows, and the variance of the marginal densities of internal and external states. The examination of a categorical system demonstrated that this is not some peculiarity of linear Gaussian stochastic processes, but instead that the transient violation of the conditional independencies of a Markov blanket applies more generally once initial conditions are imposed.

### 7.2. Memory in Cognitive Systems

In the cognitive sciences, memory is subclassified into different types [42]. The form of memory illustrated in this paper is generic, in the sense that all forms of memory in cognitive systems must involve transient deviations from a Markov blanket—even if some of those transients persist for many years. However, it is interesting to consider what additional conditions we would need to impose to interpret the analysis here in terms of different forms of cognitive memory.

The first distinction normally encountered in memory research is that between short-term and long-term memory. This implicates the diffusion tensor which, when small, delays return to the steady state density; and the variance of the steady state density which, when large, permits sustained deviations following perturbation. While the duration of a memory appears to be on a continuum, based upon the analysis above, there are biological considerations that favor the separation of time scales found in psychology. Specifically, populations of neurons have time constants, in the order of microseconds to hundreds of milliseconds [43], that are much shorter than the time constants associated with changes in synaptic efficacy, in the order of minutes to hours [44]—although there is considerable variation within each of these categories. Note, for example, the relatively small fluctuations in the average trajectory plot in Figure 1, which is considerably smaller than the fluctuations in the trajectories over which this average was constructed. This implies smaller diffusion constants, and longer timescales, can be obtained by taking summaries of large numbers of neurons or synapses. The slowing associated with this sort of summarization underwrites the renormalization-group approach that has been used to identify a range of timescales associated with neural processing [45]. In this approach, the steady-state density—or solution to density dynamics—pertains to the timescale in question, where coarse-graining operators allow for the steady-state density itself to change at a slower timescale. The analysis presented in this paper is agnostic to timescale, as we have not equipped the temporal axis with units. However, if we were to impose units of seconds, the simulations presented above might be equally applicable to a system whose potential (negative log steady state density) is time invariant and to a system whose potential changes with a time constant of hours or days (i.e., that is approximately static over the course of a few seconds).

Within short-term memory there are many subcategories. Of these, working memory is perhaps the most prominent. This is defined as memory that persists over seconds in which a memory is held ‘online’ for the purposes of some cognitive processing (e.g., for planning) [46]. It is often associated with structures like the prefrontal cortex [8,43,47], where neural activity evolves over longer durations than in sensory cortical regions [48]. The inclusion of a behavioral aspect in definitions of working memory hints at why it is useful to maintain direct conditional dependencies between internal and external states. The experiments designed to probe working memory offer a concrete way of expressing this. A typical set-up [49,50,51,52,53] starts by presenting an experimental participant with a stimulus. This may be one of several possible images, sounds, or an indicator for a location on the stimulus display. The stimulus is then removed and, following a delay, a behavioral response is solicited from the participant. Crucially, the correct response depends upon the initial stimulus. For instance, the participant may be asked to identify the original image from a selection of possible images, or to perform a saccadic eye movement to the location indicated by the original stimulus. 

In this set-up, the external state is the correct response. The blanket states include the behavioral response and the stimulus presented. The internal states are the neural states involved in causing the behavioral response. The presentation of the stimulus at the start of the trial represents an imposition of initial conditions. Accurate performance of the task requires maintenance of conditional dependencies between external and internal states throughout the delay period, such that internal states guide the appropriate behavioral response. Although the correct response stays the same for the trial in question, if we were to project our beliefs forwards to the next trial, the initial conditions for the current trial are irrelevant, and the Markov blanket is restored. Note that the set-up described here takes place in the categorical domain. Similarly, computational models of working memory tasks are sometimes formulated in terms of categorical partially observed Markov decision processes [54,55,56]. This underwrites the importance of demonstrating the emergence of conditional dependencies in Markov blanketed categorical systems.

Long-term memory is often divided into declarative (explicit) and non-declarative (implicit) forms [57,58]. Loosely speaking, the former are memories that can be communicated (i.e., declared) and include episodic and semantic memories. The latter include things like procedural memories. Procedural memories are learned patterns of behavior [59], implicating structures like the cerebellum, supplementary motor cortices, and basal ganglia [60,61,62]. For example, learned reaching behavior [63] involves a gradual synchronization, over repeated imposition of new initial conditions, between external states representing trajectories of joint angles and the internal states representing trajectories of neural states (e.g., the pattern of synaptic weights determining sequential activation of motor neurons [64,65]). The move towards Markov blanketed variables representing parameters of trajectories may be important for understanding more temporally structured mnemonic dynamics [10].

The declarative memory category includes episodic and semantic memories. The main difference between the two is that episodic memories are tied to the spatiotemporal context in which the blanket states causing the memory occurred, while semantic memories are divorced from that context. The implication is that the internal and external states include time and space variables. Neural manifestations of these might include hippocampal place and time cells [66,67]. Episodic memories then involve the emergence of conditional dependencies across the blanket between internal states representing the association between time, space, and the event causing changes in the blanket, and external states representing the same association. For instance, if navigating a maze, we might consider our location following each step we take. This could be parameterized by a random variable (or set of variables) that represent the transitions between each location from one step to the next. The corresponding internal states might be the synaptic efficacies connecting a sequence of place cells. Slow transient conditional synchronization between these efficacies and the transitions from one step to the next would represent a primitive form of episodic memory, as the synaptic efficacies would retain information about the spatiotemporal trajectory through the maze. Semantic memories may include similar associations, but with variables other than time and space. In place of external variables being alternative transitions—associating a location at one time with that at the next—they may associate quantities, like the title of a book and its author, that do not have a spatiotemporal structure.

In summary, long-term memory and short-term memory can be disambiguated from one another based upon the time taken for loss of conditional dependencies between internal and external states, reflected by the time constants associated with their physiological substrates. Subcategorization of these temporal categories is largely based upon what the internal, external, and blanket states are. As such, the abstract treatment in this paper is limited in its potential contribution to this level of classification. To take this further, we would have to commit to specific identities for each set of states for the context we hoped to understand.

### 7.3. Limitations

The numerical simulations above are designed to provide simple proofs of principle, but do not represent an exhaustive analysis of the relationships between different parts of the equations of motion. In addition, the numerical examples are limited to systems whose probability distributions are parameterized such that the flows of the sufficient statistics are given by linear functions. This may be appropriate in assessing regions close to the fixed points for these variables when the flow is non-linear but is more limited in understanding the behavior of a system more globally.

Another limitation is that we have used examples in which the dimension of each component of the system is one. The limitations due to this choice are threefold. First, this limits the form of the dynamics intrinsic to internal, external, or blanket states. Higher dimensions might lead to more complex (e.g., periodic) dynamics within each component that could have implications for the kinds of synchronization that could develop between components. Secondly, deep temporal models [68,69,70] of the sort used in theoretical neurobiology depend upon their being multiple dimensions of internal state that induce a separation of time scales. Finally, for application to biological systems with an action-perception cycle [71], it is often necessary to divide blanket states into active and sensory partitions [10,72], implying the blanket states must be at least two-dimensional. 

## 8. Conclusions

The main ideas in this paper have been formulated as a pre-emptive resolution of an apparent paradox; namely, a steady-state density with a Markov blanket precludes conditional dependencies that could be read as ‘forgetting’ or ‘memory’. These pre-emptive arguments—based upon density dynamics from initial conditions—provide a useful device with which to unpack some interesting aspects of Markov blanketed stochastic systems and disclose some interesting behaviors that rest on transient violations of blanket conditions. Memory is one example of this—formulated here as a transient synchronization across a Markov blanket.

## Figures and Tables

**Figure 1 entropy-23-01105-f001:**
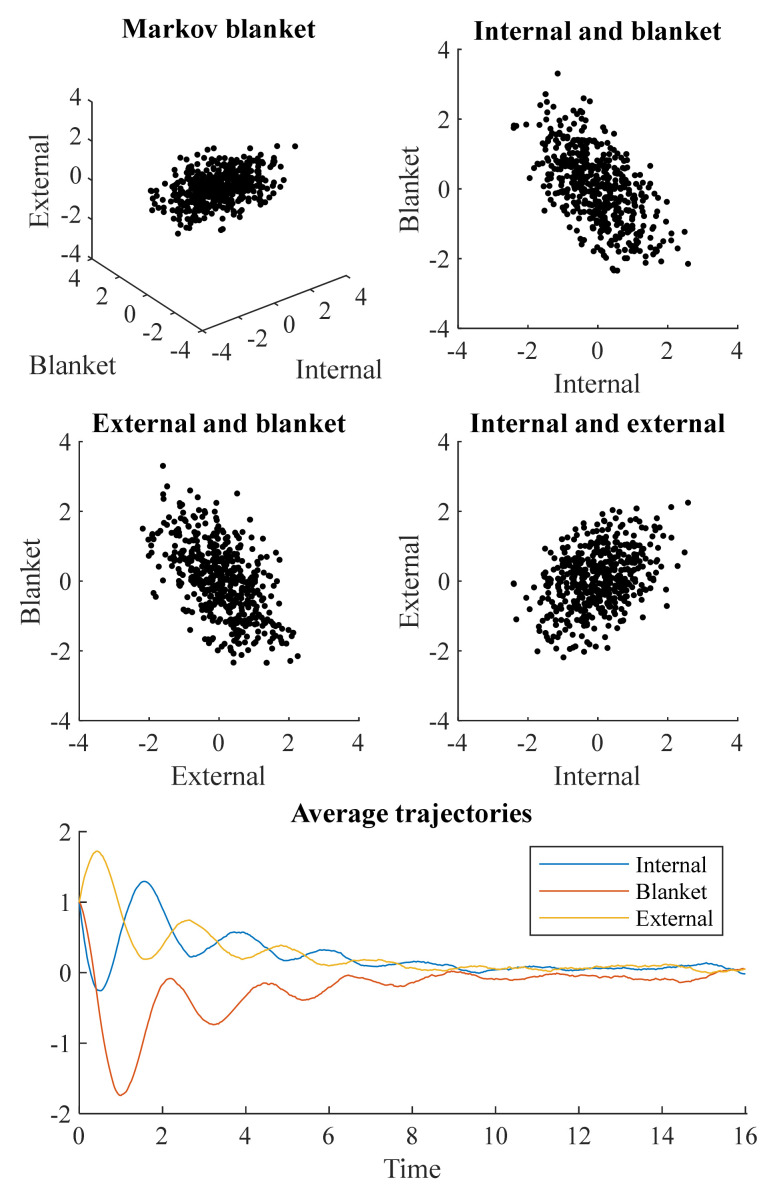
Stochastic evolution to a Markov blanket. This figure illustrates the evolution of a stochastic differential equation towards a steady state density with a Markov blanket. The upper four plots show the final state of 512 simulations at *τ* = 16. Note the negative correlation between the internal and blanket states (*ρ* = −0.53), and between the external and blanket states (*ρ* = −0.55). These result in a positive correlation (*ρ* = 0.37) between internal and external states, despite the two being conditionally independent given blanket states. The lower plot shows the average value of each state (averaged over simulations). The overall picture is that of a damped oscillatory process. Time is expressed in arbitrary units.

**Figure 2 entropy-23-01105-f002:**
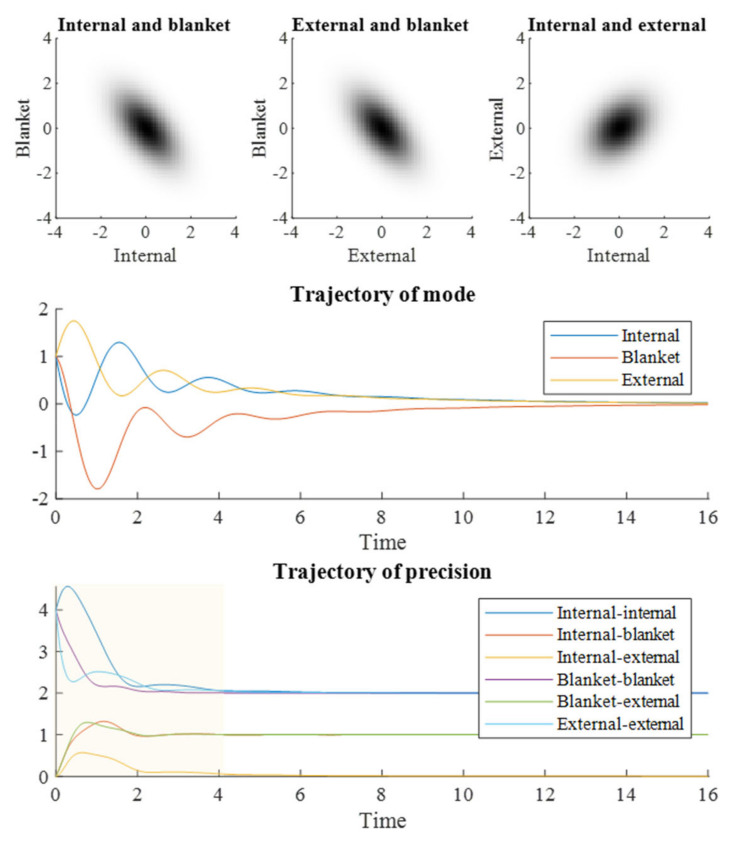
Density dynamics. This figure shows a re-expression of the results in Figure 1 through solving a (Laplace) parameterized version of the associated Fokker–Planck equation. Note that the Laplace assumption holds exactly for the (Ornstein–Uhlenbeck) process simulated here. The upper row of plots shows the densities at the end of the simulation. The middle plot shows the mode of the multivariate normal distribution as it evolves over time, and the lower plot shows the elements of the precision (inverse covariance) matrix during this evolution. The interesting observation from this figure is that the internal and external states acquire a transient synchronization or, more precisely, a conditional dependence upon one another. This is evident in the non-zero element of the precision matrix linking the two during the shaded part of the plot, which then decays back to zero. Time is expressed in arbitrary units.

**Figure 3 entropy-23-01105-f003:**
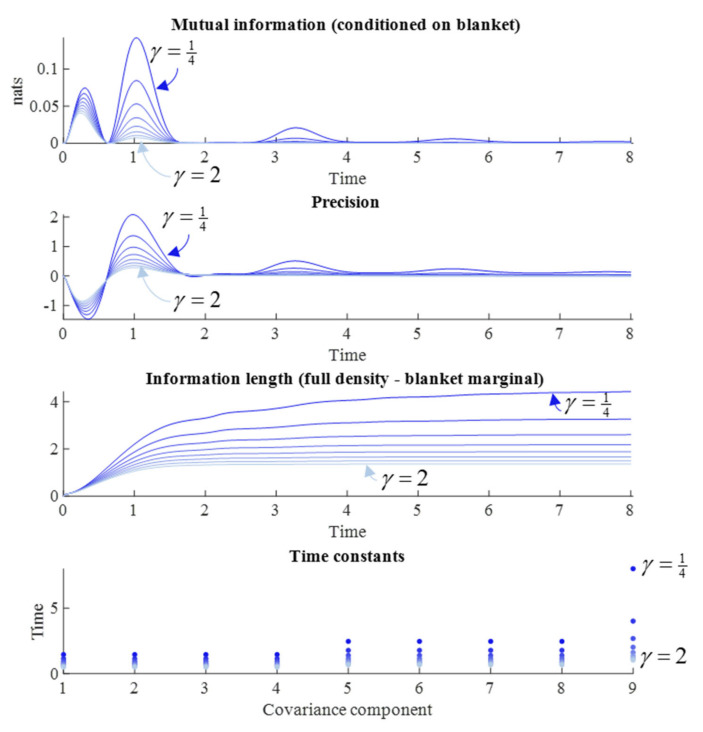
Dissipative flows and measures of memory. This figure shows the influence of the rate constants for the dissipative flow of the internal and external states on different measures of memory. These are changed symmetrically, starting from a value of *γ* = ¼ (darkest blue), and increasing this in unform intervals until *γ* = 2 (lightest blue). This range was chosen to capture the interesting behavioral spectrum of the system, while ensuring all simulations could be clearly shown in the same plot. However, there is nothing special about these limits. They can be customized at the user’s discretion (in the code associated with this paper: Please see the Data Availability Statement for details). The first plot shows the evolution of the conditional mutual information between internal and external states over time. The second shows the element of the precision matrix linking the internal and external states. The third shows the difference between the information length travelled by the joint density and the blanket marginal. The fourth shows the reciprocals of the absolute values of the real parts of the eigenvalues associated with the Jacobian for the elements of the covariance matrix. Given that the real parts are negative or zero, they play the role of decay constants for each eigenvector of the Jacobian. For a given eigenvalue (*λ*) the ratio of the amplitude of the associated eigenvector at the start and end of a time interval Δ*τ* is exp(*λ*Δ*τ*), meaning the absolute value of the reciprocal of *λ* is equivalent to the time taken for the amplitude of the eigenvector to decrease to 1/e of its initial value. Time is expressed here in arbitrary units.

**Figure 4 entropy-23-01105-f004:**
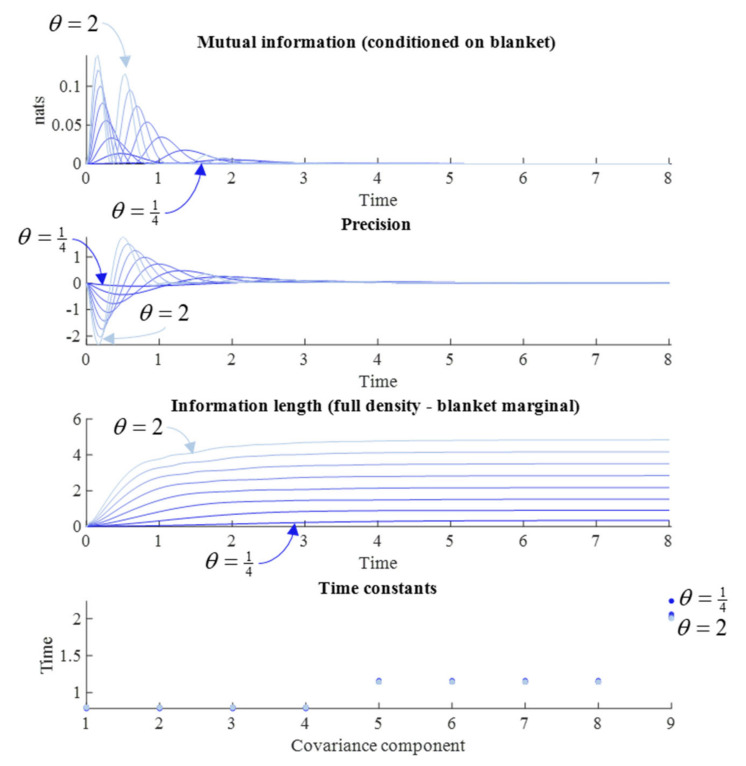
Solenoidal flow and memory. This figure uses the same format as Figure 3 but keeps *γ* constant at one—which sits in the middle of the range used in Figure 3—and varies from *θ* = ¼ (darkest blue) to *θ* = 2 (lightest blue), again using uniformly increasing intervals. Time is expressed in arbitrary units.

**Figure 5 entropy-23-01105-f005:**
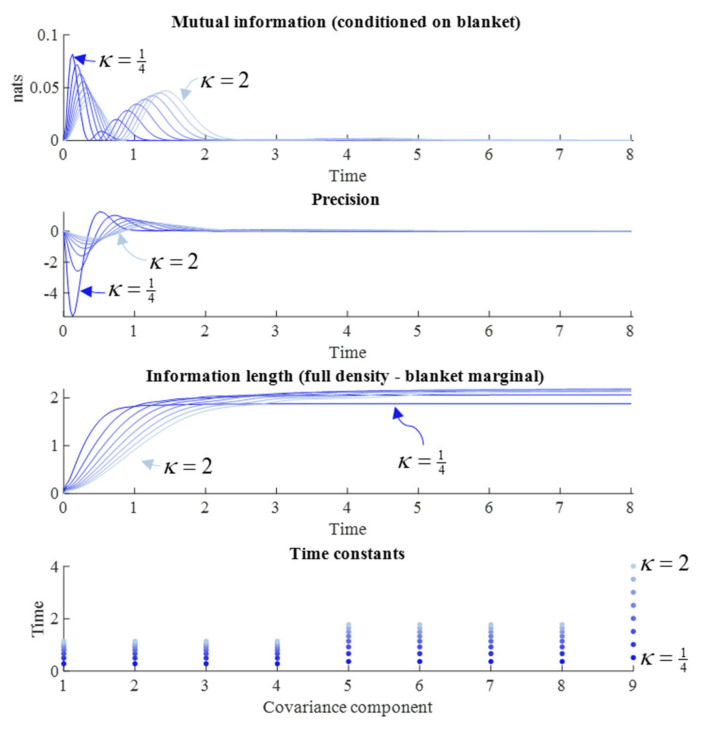
NESS variance and memory. This figure uses the same format as Figure 3 but keeps *γ* constant at one and varies from *κ* = ¼ (darkest blue) to *κ* = 2 (lightest blue), again using uniformly increasing intervals. Time is expressed in arbitrary units.

**Figure 6 entropy-23-01105-f006:**
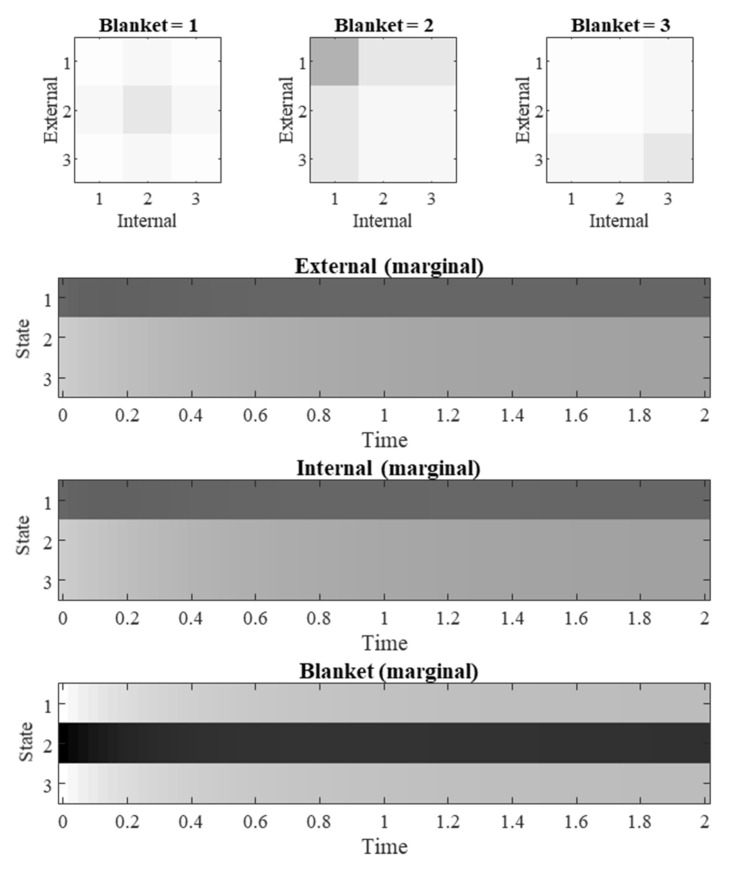
Categorical probabilistic dynamics. These plots show the evolution of the system specified in Table 3, where *γ* = *κ* = *θ* = 1. The upper row of plots depicts the probability at *τ* = 2 for each combination of internal, external, and blanket state. The evolution of the marginal distributions of each state are shown in the lower plots. Each row represents a level that a state can occupy. The shading indicates the probability of occupancy of that level, with white indicating zero, black indicating one, and intermediate probabilities by shades of grey. Time is expressed in arbitrary units.

**Figure 7 entropy-23-01105-f007:**
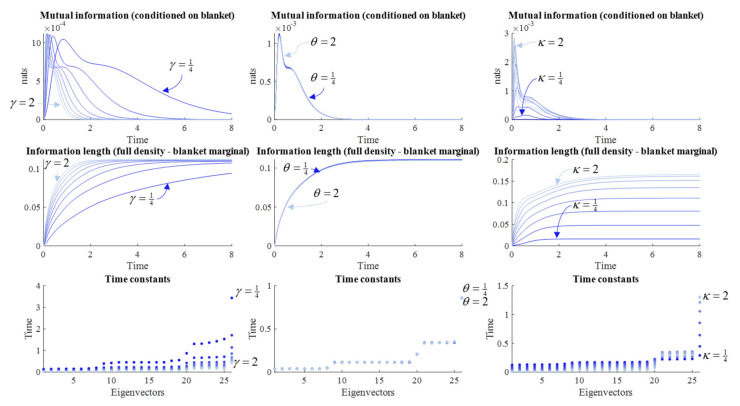
Memory in categorical systems. These plots show the same analysis as Figure 3, Figure 4 and Figure 5, but for categorical systems, and for the parameters as specified in Table 3. The plots on the left report the effect of changing *γ*, the middle plots report the effect of changing *θ*, and the right plots report the effect of changing *κ*. The mutual information and information length are calculated as for the continuous system, although the mutual information is then averaged under the distribution of blanket states to arrive at a single value. The time-constants are the negative reciprocals of the probability transition rate matrix (**L**). Time is expressed in arbitrary units.

**Table 1 entropy-23-01105-t001:** Parameters of stochastic system. This table sets out the parameterization of the stochastic system used for the simulations.

Parameter	Value
ϑ	$\left[\begin{array}{l} 0\\ 0\\ 0 \end{array}\right]$
Q	$\left[\begin{array}{ccc} 0 & -\theta & 0\\ \theta & 0 & -\theta \\ 0 & \theta & 0 \end{array}\right]$
Π	$\frac{1}{\kappa }\left[\begin{array}{ccc} 2 & \kappa^{\frac{1}{2}} & 0\\ \kappa^{\frac{1}{2}} & 2\kappa & \kappa^{\frac{1}{2}}\\ 0 & \kappa^{\frac{1}{2}} & 2 \end{array}\right]$
Γ	$\frac{1}{4}\left[\begin{array}{ccc} \gamma & 0 & 0\\ 0 & 1 & 0\\ 0 & 0 & \gamma \end{array}\right]$

**Table 2 entropy-23-01105-t002:** Gaussian conditionals and marginals. This table reproduces standard results [25] for the forms of conditional and marginal densities of a multivariate Gaussian distribution for ease of reference.

Distribution	Mode	Covariance
p(α,β)	$\vartheta_{\alpha },\vartheta_{\beta }$	$\left[\begin{array}{cc} \Sigma_{\alpha \alpha } & \Sigma_{\alpha \beta }\\ \Sigma_{\alpha \beta }^{T} & \Sigma_{\beta \beta } \end{array}\right]$
p(α|β)	$\vartheta_{\alpha }+\Sigma_{\alpha \beta }\Sigma_{\beta \beta }^{-1}(\beta -\vartheta_{\beta })$	$\Sigma_{\alpha \alpha }-\Sigma_{\alpha \beta }\Sigma_{\beta \beta }^{-1}\Sigma_{\alpha \beta }^{T}$
p(α)	$\vartheta_{\alpha }$	$\Sigma_{\alpha \alpha }$

**Table 3 entropy-23-01105-t003:** Parameters of categorical system. This table sets out the parameterization of the stochastic system used for the simulations. The *σ* symbol indicates a softmax (normalized exponential) function.

Parameter	Value
$s_{b}(\infty )$	$\sigma \left(\kappa \ln\left(\left[\begin{array}{l} 1\\ 3\\ 1 \end{array}\right]\right)\right)$
$s_{\eta |b}(\infty )=s_{\mu |b}(\infty )$	$\sigma \left(\kappa \ln\left(\left[\begin{array}{ccc} 1 & 3 & 1\\ 3 & 1 & 1\\ 1 & 1 & 3 \end{array}\right]\right)\right)$
Γ	$\frac{\gamma }{128}\left[\begin{array}{cccc} -26 & 1 & 1 & \cdots \\ 1 & -26 & 1 & \\ 1 & 1 & -26 & \\ \vdots & & & \ddots \end{array}\right]$
Q	$\frac{\theta }{512}\left[\begin{array}{ccc} 0 & -1 & 1\\ 1 & 0 & -1\\ -1 & 1 & 0 \end{array}\right]⊗\left[\begin{array}{ccc} 0 & -1 & 1\\ 1 & 0 & -1\\ -1 & 1 & 0 \end{array}\right]⊗\left[\begin{array}{ccc} 0 & -1 & 1\\ 1 & 0 & -1\\ -1 & 1 & 0 \end{array}\right]$

## Data Availability

The simulations presented in this paper were generated by MATLAB code available at https://github.com/tejparr/Memory-Markov-Blankets.

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
