# Peer review of "Memory and Markov Blankets"

_entropy, 2021, doi:10.3390/e23091105_

Round 1

Reviewer 1 Report

Summary: Random dynamical systems (brain, etc ) that appear to model their environments, can be represented by appealing to the existence of a steady-state, whose density preserves conditional independence between a biological entity and its surroundings. From this perspective, the conditioning set, or Markov blanket, induces a form of synchrony between creature and world as if one were modeling the other. However, this results in an apparent paradox. If all conditional dependencies between a system and its surroundings depend upon the blanket, how do we account for the mnemonic capacity of living systems? It might appear that any shared dependence upon past blanket states violates the independence condition, as the variables on either side of the blanket now share information not available from the current blanket state. This paper aims to resolve this paradox, and to demonstrate that conditional independence does not preclude memory.  The interesting question then becomes: what determines the length of time required for a stochastic system to ‘forget’ its initial conditions? The authors explore this question for a system, whose steady state density possesses a Markov blanket, through simple numerical analyses. 

It seems the problem presented by the authors could be further explored, for this, I raise some questions that seek to complement the material presented. 

A) About the role of the diffusion tensor:

pag 11, line 341: the authors state:
*We might anticipate that systems with fast fluctuations should very rapidly forget their initial conditions, such that large γ should lead to rapid forgetting.*

(1) Describe in a simple way (equation or part of) how fast are the fluctuations in connection with the values of γ.
(2) The values γ=1/4 and γ=2 are limits in some sense? Please, explain how those 2 values are identified.
(3) It is possible to estimate the ideal value of γ, needed to describe a real system (in a lapse of time)?   

I know that figure 3 shows this, but it is hard to qualify the differences. To provide an accurate notion of the differences (under the variations of γ), then I suggest including some pieces of information displayed in a table. 

B) 

Pag 13, the caption of figure 4, the authors state:
*This figure uses the same format as Figure 3 but keeps γ constant at one and varies from θ = ¼ (darkest blue) to θ = 2 (lightest blue), again...*

(1) Taking into account the content of section 5.1., what is the sense of using γ = 1? 
(2) How is the behavior of figure 4 for large/small range values of γ? 
(3) It is possible to estimate the ideal value of θ, needed to describe a real system (in a lapse of time)?  

C) 
Note those equivalent observations of B) arise for figure 5, in the caption the authors state: 
*This figure uses the same format as Figure 3 but keeps γ constant at one and varies from κ = ¼ (darkest blue) to κ = 2 (lightest blue), again*

Note also that κ has a different role when compared with γ and θ, in terms of the inferential perspective.
Under a Bayesian perspective, K can be understood as a hyperparameter, since it is associated with the hyperparameters of equation (13). Then, how it can be defined in practice?

D) Sections 5 and 6 could be closed with a couple of applications to the real world, which would allow confronting the challenges of representation, which seems to be one of the proposals of this paper. For instance, the authors mention some possibilities on line 602 of page 20. 

Reviewer 2 Report

See attached file.

Round 2

Reviewer 2 Report

The authors have complied with my concerns about the paper. Editors can accept it in its present form.

Author Response

Many thanks for this evaluation, and for your help in improving the manuscript